# Analysing the Impact of Resistant Starch Formation in Basmati Rice Products: Exploring Associations with Blood Glucose and Lipid Profiles across Various Cooking and Storage Conditions In Vivo

**DOI:** 10.3390/foods13111669

**Published:** 2024-05-27

**Authors:** Prabhjot Kaur, Harpreet Kaur, Renuka Aggarwal, Kiran Bains, Amrit Kaur Mahal, Lachhman Das Singla, Kuldeep Gupta

**Affiliations:** 1Department of Food and Nutrition, Punjab Agricultural University, Ludhiana 141004, Punjab, Indiarenukaaggarwal@pau.edu (R.A.);; 2Department of Mathematics, Statistics and Physics, Punjab Agricultural University, Ludhiana 141004, Punjab, India; akmahal@pau.edu; 3Department of Parasitology, Guru Angad Dev Veterinary and Animal Sciences University, Ludhiana 141004, Punjab, India; 4Department of Veterinary Pathology, Guru Angad Dev Veterinary and Animal Sciences University, Ludhiana 141004, Punjab, India

**Keywords:** glycaemic index, glycaemic load, resistant starch, basmati rice, dietary fibre, cooking methods, storage temperature

## Abstract

Common cooking methods were used to prepare basmati rice products, including boiling 1 (boiling by absorption), boiling 2 (boiling in extra amount of water), frying, and pressure cooking. The cooked rice was held at various temperatures and times as follows: it was made fresh (T1), kept at room temperature (20–22 °C) for 24 h (T2), kept at 4 °C for 24 h (T3), and then reheated after being kept at 4 °C for 24 h (T4). The proximate composition, total dietary fibre, resistant starch (RS), and in vitro starch digestion rate of products were examined. The effect of RS on blood glucose and lipid profiles was measured in humans and rats, including a histopathological study of the liver and pancreas in rats. The basmati rice that was prepared via boiling 1 and stored with T3 was found to be low in glycaemic index and glycaemic load, and to be high in resistant starch. Similarly, in rats, the blood glucose level, cholesterol, triglycerides, and LDL were reduced by about 29.7%, 37.9%, 31.3%, and 30.5%, respectively, after the consumption of basmati rice that was prepared via boiling 1 and stored with T3. Awareness should be raised among people about the health benefits of resistant starch consumption and the right way of cooking.

## 1. Introduction

The development of the latest handy food products and the use of new technologies have both improved in current years [1]. The recognition of the connection between a nutritious food regimen and good health is one of the important reasons for the popularity of functional foods, which provide health advantages and further lessen the chance of persistent sicknesses due to fundamental vitamins. Resistant starch (RS), which has been identified as a type of dietary fibre, is one of the most broadly used ingredients in functional foods. It has long been recognised for having a superb function among fibres regarding its several nutritional features that provide health benefits. Its consumption lowers the blood glucose levels after a meal, reduces cholesterol and triglyceride levels in blood, improves the body’s response to insulin, and even reduces fat storage [2]. Animal studies have shown that resistant starch has positive effects on gut health by helping to create a substance called butyrate. This can change the composition and balance of microorganisms in the gut, which is the main reason of resistant starch becoming more popular as a food ingredient [3].

Resistant starch was discovered in the 1980s as a component that resists digestion by enzymes [4], and it has been defined as ’the sum of starch and products of starch degradation no longer absorbed in the small intestine of healthy individuals’ [5]. Foods contain five main types of resistant starch. Resistant starch 1 (RS1) is a type of starch that the body cannot digest because of the presence of whole, undamaged cell walls on grains, tubers, and seeds. This is used as an ingredient in a vast range of traditional foods due to its stability to heat during normal cooking processes [6]. Because of their crystallinity, RS2 are natural, uncooked starch granules that are not greatly affected by hydrolysis. RS2 is found in raw starch granules like banana, potato, and high amylose corn starches. RS3 is retrograded starch, which is created during cooking and stored at room temperature or below [7]. RS4 starches include cross-linked and starch ethers, which are chemically modified to develop resistance against enzymatic digestion [8]. Amylose–lipid complexes, which occur when amylose and lipids come into contact, are where RS5 is created. Amylose–lipid complexes are created when lipids perforate the amylose chains found in numerous plant sources with high amylose concentrations [9].

The mechanism that causes the starch chains to recrystallise after the gelatinised paste cools is known as retrogradation. The extent and rate of the retrogradation of starch are determined by its features, such as its crystalline or molecular structure, and storage circumstances, such as duration, temperature, and water content [10].

The delayed digestion of resistant starch, which takes more than five to seven hours, results in the reduction of insulinemia and postprandial levels, and may lengthen the duration of satiety [11]. Its physiological actions are primarily responsible for its remarkable nutritional activity when compared to dietary fibres. When subclasses of fermentable dietary fibre sources are regularly consumed, there is a large synthesis of short-chain fatty acids (SCFAs) and gut-related microbiota, and immunological regulation occurred [9]. Resistant starch avoids the small intestine’s digestive process and passes straight into the large intestine, where it ferments into short-chain fatty acids like acetate, propionate, and butyrate, as well as gases like H_2_, CO_2_, and CH_4_ that have been produced by the large intestine’s probiotic bacteria [12]. The SCFAs are absorbed via the intestinal wall and transported into the liver via portal circulation, where mainly butyrate gets utilised by colonocytes [13]. The SCFAs protect the human body from metabolic disorders like obesity by providing energy to our brain muscles and heart. It also increases the production of bile acids, mineral absorption, and leptin production [14]. Consuming resistant starch promotes the growth of beneficial bacteria like Lactobacillus, Bacteroides, and Bifidobacteria, while inhibiting the growth of harmful bacteria like Firmicutes [3]. Several methods have been described by RS to lower cholesterol levels, including the increased excretion of bile in the faeces and a reduction in the influence of propionic acid on cholesterol production. In rats, SCFSs also lessen the synthesis of cholesterol in the gut and liver [15]. Hence, it is suggested that at least 20 g of resistant starch per day should be consumed to obtain the various health benefits. In developing countries, about 30 to 40 g/d of resistant starch is recommended [16], while in India and China, RS consumption of about 10 and 18 g/d, respectively, is recommended, indicating that about 10 to 20% of daily carbohydrate consumption in the form of resistant starch is necessary for the health benefits to be obtained [17].

The cooking and grinding of food products for a longer period of time reduces its resistant starch content. During grinding, the resistant starch content of rice and oats is reduced from 12 to 5% and from 16 to 3% respectively. Resistant starch formation in food is also affected by water content, pH, heating processes (time, temperature, etc.), preservation methods (freezing, drying, storage, etc.), cooking methods, and the presence of components such as proteins, lipids, minerals, and inhibitors [6]. Resistant starch content has been found to be impacted by both dry heating techniques, including baking, frying, microwaving, and autoclaving, as well as wet cooking techniques, like pressure cooking, steaming, and boiling. A high baking temperature and time has been reported to increase the resistant starch content [18]. 

The main challenge within the food industry is the manufacturing of consumer-friendly foods, which contains enough resistant starch to result in a great enhancement of public health. Though the health benefits provided by resistant starch have been well documented, few studies are available regarding the resistant starch content of basmati rice products in India, and the in vivo efficacy of the resistant starch of basmati rice products in improving glucose and lipid profiles has not been studied. Considering the health benefits of resistant starch, the current study was designed to determine the manner in which cooking, and storage temperatures affected the resistant starch in basmati rice products and how that affected the lipid profile and blood sugar levels. Dietary interventions using resistant starch may improve the glucose metabolism and insulin sensitivity. Thus, products made from basmati rice were selected because they represent an essential part of the diet of people from North India. We additionally examined how well the resistant starch in basmati rice products worked in vivo enhanced the lipid profiles and blood glucose levels in rats and humans. 

Our objectives are as follows:To determine the effect of cooking and refrigeration on the resistant starch and dietary fibre content of cereal products;To quantify the resistant starch and soluble fibre components in cereal products;To assess the impact of resistant starch and soluble fibre components on postprandial glucose responses and appetite ratings;To determine the effectiveness of resistant starch on lipid profiles and blood glucose levels in rat models.

## 2. Material and Methods

### 2.1. Procurement and Cooking

The most common form of basmati rice (PB1121) was obtained by Punjab Agricultural University in Ludhiana’s Department of Plant Breeding and Genetics This variety has been grown in large amounts in the fields of Punjab. It was harvested in the month of October from the field. The grains were cleaned and milled in order to remove the husk. The moisture content of basmati rice was reduced by up to 12 percent after milling. Four common cooking methods used by North Indians, namely boiling 1 (Boiling by absorption; rice were boiled in equal amount of water), boiling 2 (Boiled in extra amount of water), frying (Fried rice), and pressurecooking, were chosen (Table 1). These four food products were examined under four distinct storage conditions, or four treatments, as follows: they were prepared freshly (T1), kept at room temperature (20–22 °C and 45–50% RH) for 24 h (T2), stored at 4 °C for 24 h (T3), and then reheated after being kept at 4 °C for 24 h (T4) (Figure 1). Every rice product was utilised three times during the trial. The samples were dried and stored in zip lock bags (airtight packages) and further used for nutritional analysis after the treatments.

### 2.2. Nutritional Analysis 

#### 2.2.1. Proximate Composition 

Using standardised techniques, the nutritional analyses of cooked samples were performed for crude protein, crude fat, and ash. Crude protein was measured using the macro-Kjeldahl technique. The (AOAC 2000) method was also used to measure the amount of crude fat and ash [19].

#### 2.2.2. Dietary Fiber

A megazyme total dietary fibre (K-TDFR-200A) kit was used to calculate the total amount of dietary fibre. The standard methodology provided by [19] was also utilised to analyse the contents of soluble and insoluble dietary fibre. The following formula was used to determine the dietary fibre:(1)$$Dietary\;fibre(\%)=\frac{\frac{R_{i}+R_{2}}{2}-P-A-B}{\frac{m_{1}+m_{2}}{2}}\times 100$$
where

*R*_1_ = residue weight 1 from m_1_, *R*_2_ = residue weight 2 from m_2_, *m*_1_ = sample weight 1,

*m*_2_ = sample weight 2, *A* = ash weight from *R*_1_, *p* = protein weight from *R*_2,_ and
(2)$$B=blank=\;\frac{BR_{1}+BR_{2}}{2}-BP-BA$$
where

*BR* = blank residue, BP = blank protein from *BR*_1_, BA = blank ash from *BR*_2_.

#### 2.2.3. Total Starch and Resistant Starch

Using a megazyme K-RSTAR assay kit provided by [20], the total and resistant starches were calculated. To find the total amount of starch, non-resistant (solubilised) starch and resistant starch were added.

#### 2.2.4. In Vitro Starch Digestion Rate

The method described by [21] for determining the in vitro starch digestion rate was used. Furthermore, 500 mg of the sample was incubated with 250 U porcine amylase in 1 mL of artificial saliva (carbonate buffer; Sigma A-3176 Type VI-B) [3050 spruce, St. Louis, MO 63103 USA]) for 15 to 20 s. A water bath at 37 °C was used to incubate 5 mL of pepsin (1 mL per ml of 0.02 M aq. HCl; from gastric porcine mucosa; Sigma P-6887) for 30 min. Before correcting the pH to 6, the digesta was neutralised by adding 5 mL of 0.02 M aq. Sodium hydroxide. (0.2 M C_2_H_3_NaO_2_ buffer, 25 mL). Two additional ingredients were added as follows: 2 mg of pancreatin per ml of acetate buffer (Sigma P1750 from porcine pancreas) and 5 mL of amyloglucosidase (Sigma A-7420 from Aspergillus niger; 28 U per mL of acetate buffer). The solution was then incubated for 4 h, during which the digesta’s glucose concentration was periodically checked using an Accucheck (Roche Diabetes Care, India) glucometer.

#### 2.2.5. Rapidly Digestible Starch and Slowly Digestible Starch

The following formula was used to translate the glucometer reading at 15 min to the percentage of starch digested.
DS = (0.9 × G_G_ × 180 × V)/(W × S [100 − M])(3)
where

G_G_ = Reading of the glucometer (mM/L);

V = Digest volume (mL);

180 = glucose’s molecular weight;

W = sample weight (g);

S = sample’s starch content (g per 100 g dry sample);

M = moisture percentage in sample (g per 100 g sample);

0.9 = starch stoichiometric constant from glucose concentrations;

RDS% = percentage of starch digested at 15 min;

SDS% = percentage of starch digested at 120 min-percentage of starch digested at 15 min.

### 2.3. Impact of Resistant Starch and Soluble Fibre Components on Postprandial Glucose Response via Measuring Glycaemic Index

The formula for the glycaemic index was computed by applying Goni’s approach [22]. All subjects provided their informed consent for inclusion before they participated in the study. The protocol for the study was approved by the ethics committee of Punjab Agricultural University, Ludhiana. The blood glucose levels were measured in ten healthy participants. In accordance with the Helsinki Declarations, the participants provided their explicit written consent. The participants were instructed to maintain a 12 h fast over night. The test food was served in the morning, and participants had fifteen minutes to eat their meals. A finger prick was used to obtain blood samples using a Glucometer (Dr. Morphine). The blood glucose level was assessed at 0, 15, 30, 45, 60, 90, and 120 min following the ingestion of 50 g of cooked cereal product, which is equivalent to a fast. Furthermore, 50 g of glucose were provided to the control group in order to compare how cooked food affected their blood sugar levels. Then, 150–300 mL of water could be consumed by volunteers, depending on what they ate throughout the study. The following formula was used to determine the *glycaemic index*:$$GI=\frac{Area\;under\;the\;curve\;for\;50\;gm\;carbohydrate\;for\;test\;sample}{Area\;under\;the\;curve\;for\;50\;gm\;carbohydrates\;from\;control\;(glucose)}\times 100$$

Glycaemic load was calculated as
$$Glycemic\;load=\frac{GI\times Available\;carbohydrates}{100}$$

### 2.4. Effectiveness of Resistant Starch on Blood Glucose Levels and Lipid Profiles in Rats

It was hypothesised that consuming foods high in RS might benefit the treatment of diabetes and hyperlipidaemia, either by increasing insulin secretion, decreasing its sensitivity, or by lowering the synthesis of cholesterol. We performed a rat experiment to gather accurate, true, and unbiased data on this. Furthermore, rats and humans share a great deal of similarities in terms of biology and genetics, as well as behavioural traits.

#### 2.4.1. Animal Collection

Thirty-five Wistar albino rats, weighing between 180 and 220 g and 2 to 3 months old, were acquired from the Akal College of Pharmacy and Technical Education Mastuana sahib, Sangrur (Registered breeder of CCSEA) animal house and breeding centre (AHBC). The Institutional Animal Ethics Committee granted authorisation for the experiment to be conducted (IAEC no.: GADVASU/2023/1AEC/68/12). The animals were kept in cages, were provided with commercial pellets to eat, and had unlimited access to water.

#### 2.4.2. Induction of Diabetes

Intraperitoneal injections of freshly produced 230 mg/kg nicotinamide (NA) with buffer saline NaCl 0.9% were administered to the Wistar albino rats. The rats were again provided with an intraperitoneal injection of approximately 60 mg/kg of streptozotocin (STZ) after a quarter of an hour. After injection, rats were provided with 5% glucose water in order to avoid hypoglycemia. Their blood samples were obtained five days into the induction process, and the levels of lipids, insulin, and blood glucose were assessed. More than 200 mg/kg of blood glucose was a sign of diabetes in the rats. The blood glucose levels were monitored during the first, third, and last weeks of the 28-day treatment period for the rats. The first and concluding weeks of the trial were devoted to measuring the lipid profile and insulin.

#### 2.4.3. Treatment Protocol

Group-I: (Normal control) Normal rats were fed a normal diet for a period of 28 days.Group-II: (Diabetic control) Diabetic rats were fed a regular diet for 28 days following the onset of diabetes.Group-III Diabetic rats were fed an FBR (freshly prepared boiled basmati rice) diet for 28 days.Group-IV (Treatment group) Diabetic rats were fed 4BR (stored at 4 °C for 24 h boiled basmati rice) for 28 days.Group-V (Treatment group) Diabetic rats were fed RehBR (reheated basmati rice after being stored at 4 °C for 24 h) for 28 days.

Each group contained seven rats. Among the four different types of cooking, only rice prepared via the boiling 1 method were fed to the rats as a limited numbers of rats were available to us. Moreover, the boiling 1 method is the most common way of cooking of rice among the north Indian population, and rice prepared via this method was found to have a high amount of resistant starch content and a lower glycaemia index than any other cooking method. The rice that was kept at room temperature for the entire day (T2) was also removed from the treatment since the product developed microorganisms as a result of the storage.

### 2.5. Histopathological Study of Rat Organs

#### 2.5.1. Preparation of Samples

After the termination of the experiments, the pancreas and livers were examined for histopathological studies. These organs were cut into thin-sectioned pieces (1 mm × 1 mm × 1 mm). The samples of tissue were gathered and preserved with 10% formalin. After that, the samples were placed in alcohol at successive concentrations—70%, 80%, 95%, and pure alcohol—in order to extract the water from the tissue. Subsequently, the material was purified using xylol and then embedded in block paraffin. Next, a microtome was used to cut the paraffin blocks into 5 μm thick sections in order to prepare the tissue for sectioning. The tissue portion was then left on the hot plate set at 50 °C for 15 min [23].

#### 2.5.2. Haematoxylin–Eosin (HE) Staining

Haematoxylin–eosin staining was used in a number of procedures. The tissue sample was first deparaffinised by dipping the sample into xylol I, xylol II, and xylol III for three minutes each. Subsequently, the tissue samples underwent rehydration with the addition of ethanol concentrations in increments of 100%, 95%, 80%, and 70% for a duration of two minutes each. After soaking the samples in Harri’s Haematoxylin for ten minutes, they were washed for ten minutes with tap water. Additionally, the samples were submerged in eosin for ten minutes before being serially dehydrated with ethanol concentrations ranging from 70% to 100%. Xylol I, II, and III were used to hold the samples during the cleaning procedure. Following the colouring procedure, Canada balsam glue was dripped, covered in a glass cover, and then allowed to dry. All organs were then carefully investigated under a microscope (SZX16 Olympus, Ambala, India) using 600× magnification [23,24].

### 2.6. Statistical Analysis

The data were analysed using descriptive statistics, analysis of variance, and post hoc tests with SAS (version 9.4) software. A factorial completely randomised design was used to examine the quantity and effect of the cooking method and storage temperature on the proximate composition, total starch, and dietary fibre content of cereal products. One way analysis of variance was used to compare the average glycaemic index, glycaemic load, and pre-, during, and post-treatments in rat blood glucose levels. *T*-Test analysis was performed to examine the average effect of resistant starch-rich products on the insulin and lipid profiles of rats.

## 3. Results

### 3.1. Proximate Composition

It was discovered that basmati rice had the highest crude protein level (9.54%), which was prepared using the boiling 1 method, followed by frying (fried rice, 8.28%), boiling 2 (8.22%), and pressure cooking (7.49%) (Table 2). The protein content of the products held at different temperatures showed a significant variation (≤0.001*), with T3 having the greatest content (9.47%), followed by T2 (8.32%), T1 (8.03%), and T4 (7.39%). When compared to protein, the crude fat content of fried rice (10.86%) and the boiled 1 method (2.18%) was found to be higher than that of the other cooked products due to the additional oil used during the frying process. Out of all the storage temperatures, T3 had the greatest crude fat content (4.03%), followed by T2 (4%), T4 (3.92%), and T1 (3.78%). The ash content of basmati rice was found to be the highest in the boiling 1 method (1.86%), followed by frying (1.85%), boiling 2 (1.75%), and pressure cooking (1.46%). There was no noticeable difference in the ash concentration (≤0.001*) between the rice products stored at various temperatures.

### 3.2. Dietary Fibre

The soluble dietary fibre content of basmati rice was found to be higher in pressure cooked rice (0.92%), followed by boiling 2 (0.89%), boiling 1 (0.87%), and frying (0.78%). Among the storage conditions, it was found to be higher in freshly prepared food i.e., more so in T1 (1.08%) and less in T3 (0.61%). The insoluble dietary fibre was highest in the boiling 1 method (2.80%), followed by boiling 2 (2.47%), pressure cooking (T4), and frying (2.25%). It was found to be higher in T3 (2.88%) and lower in T1 (2.14%) among the different storage temperatures. The amount of dietary fibre in the prepared food products was impacted by temperature changes during storage. The amount of insoluble and total dietary fibre in basmati rice increased with the length of time and when stored at low temperatures. The highest amount of dietary fibre was found in products produced via the boiling 1 method (3.67%) and kept at 4 °C (T3) (3.49%) for 24 h. The highest amount of soluble fibre was found in fresh food samples (T1) (Table 3).

### 3.3. Resistant Starch, Non-Resistant Starch, and Total Starch

Rice is considered a starchy food product. The total starch content of rice products was found to range from 73.2 to 79.2%, with the boiling 1 method having the greatest level of total starch and boiling 2 having the lowest level (Figure 2). The RS content of raw basmati rice was found to be either increased or decreased after applying different cooking methods. Following the use of all the cooking methods, the resistant starch percentage of raw basmati rice increased to 2.27%. The resistant starch content of basmati rice was found to be highest after the boiling 1 method (12.81%), followed by frying (11.67%), boiling 2 (8.72%), and pressure cooking (7.57%). The RS content of all rice products decreased during storage at different temperatures. Products held with T3 had the highest level of resistant starch content (11.76%), followed by T2 (10.53%) and T4 (9.64%). Conversely, freshly made products, like in T1 (8.84%), showed the lowest amount of RS after cooking for 15 min at 100 °C. However, the non-resistant starch content of basmati rice was found to be higher during T1 (67.42%) when compared to T3 (63.45%), and higher during pressure cooking (67.83%) and the boiling 1 method (66.39%), respectively. The results showed that the cooking technique and the storage temperature greatly influence the structure of starch leading to the changes in its physical and nutritional characteristics.

### 3.4. In Vitro Starch Digestion Rate

The rate at which starch is metabolised in vitro is a crucial factor in assessing a food product’s ability to elevate a person’s blood glucose levels. Different storage temperatures had an impact on the in vitro starch digestion rate of basmati rice products (boiling 1, boiling 2, frying, and pressure cooking). This rate was measured 120 min after the food sample had finished digesting (Figure 3, Figure 4, Figure 5 and Figure 6). The basmati rice boiled via absorption (boiling 1) and stored with T3 and T2 had a slower digestion rate of 39 and 41% at 120 min, compared to T1 and T4 rice, which had completed digestion at a rate of 43 and 41% after 90 min in basmati rice boiled in an extra amount of water (boiling 2). The rate of starch digestion was lower in T3 and T2 products with 38 and 42.2% at 120 min as compared to T1, as well as all other rice products. The starch digestion rate of fried and pressure cooked basmati rice was also high in T1 and T4 products when compared to that stored at a low temperature. Therefore, the results indicated that the rice cooked using the boiling 1 method followed by storing with T3 had the slowest in vitro starch digestion rate at the end of the 120 min of digestion, indicating the presence of a higher amount of RS and dietary fibre in the product.

### 3.5. Rapidly Digestible Starch (RDS) and Slowly Digestible Starch (SDS)

The amount of SDS as a result of the boiling 1 method (35.37%) was highest, followed by frying (26.71%), boiling 2 (23.25%), and pressure cooking (22.03%), which is comparable to the in vitro starch digestion rate. Following T2, T4, and T1, treatment 3 (T3) raised the level of SDS to its highest point. Consequently, the SDS content is higher (Figure 7 and Figure 8). When stored with treatment 3, frying (10.71%) and boiling 1 (18.89%) were found to have the lowest RDS, followed by boiling 2 (31.49%) and pressure cooking (41.89%).

### 3.6. Impact of Resistant Starch and Soluble Fibre Components on Postprandial Glucose Response

Ten healthy human participants were fed rice products made via boiling 1, boiling 2, and pressure cooking methods that had a high RS content, testing the glycaemic response. Rice prepared using frying was not fed to the subjects as they rejected it, claiming that a fried product has a higher amount of oil. Also, basmati rice products that received treatment 2 were not offered to the subjects as they were stored at room temperature, and this temperature in the summer season led to microbial growths in the product. Moreover, in India and south-east Asia, people prefer fresh rice, but in China and north- east Asia, people prefer stale rice and use different methods to cook food. The lowest glycaemic index was observed after the consumption of basmati rice prepared via the boiling 1 method (45.8%) with T3, followed by boiling 2 (49.5%) and pressure cooked rice (50.7%). Treatment 3 was determined to have the lowest glycaemic index storage conditions, which may help those suffering from diabetes (Figure 9). Similarly, the glycaemic load was found to be high in pressure cooked rice (53.3%), followed by boiling 1 (47.2%) and boiling 2 (43.03%). But, after storing with T3, the glycaemic load was decreased, and it was found lowest in boiling 1 (34.2%) when compared to other cooking methods (Figure 10).

### 3.7. Effectiveness of Resistant Starch on Blood Glucose Level in Rats

Similar to human experiments, treatment 2 for basmati rice was not considered for rat trials. Only basmati rice prepared with the boiling 1 method were fed to rats due to the limited number available to us, and considering the fact that boiling 1 preparation has a high RS content and is a common cooking method used by the North Indian population. The findings showed that blood glucose concentrations of resistant starch from various diet groups tended to decrease, but the mean values did not differ from those of the control or diabetic control groups. In diabetic rats, the blood glucose levels increased dramatically from (114.3 ± 6.9) mg/100 mL in normal rats to (286.5 ± 16.5) mg/100 mL. Nevertheless, following the administration of the treatment diet, the pre-treatment groups’ levels dramatically restored to the normal range for each treatment group (G3–G5). After administering T1 basmati rice, the blood glucose level dropped from (286.3 ± 16.to 244.2 ± 11.0) in G3. Basmati rice stored with T3 was administered to G4, and it was discovered that the blood glucose level decreased from (282.1 ± 18.4 to 198.3 ± 18.1), and that the blood glucose level decreased from (276.6 ± 30.1 to 232.5 ± 25.8) in G5 (Table 4). The diets provided in G3, G4, and G5 resulted in A 14.7, 29.7, and 15.9% decrease in the blood glucose levels of the rats. Therefore, rice cooked with absorption techniques (boiling 1) and stored with T3 resulted in a maximum decline in the blood glucose levels of the experimental rats.

### 3.8. Effect of Resistant Starch on Lipid Profile in Rats

The diabetes group had significantly (<0.001) higher levels of triglycerides, total cholesterol, and low density lipoproteins (LDL) than the normal control and treatment groups. In contrast, the diabetic group had lower levels of high density lipoprotein (HDL). The maximum reduction in cholesterol was found in G4 with 39.9%, followed by G5 (33.5%) and G3 (32.5%), respectively (Table 5). Similarly, the triglyceride and LDL levels were reduced the most in G4 (31.3%, 30.5%), followed by G5 (21%, 17.7%) and G3 (20.4%, 11.34%), respectively. The HDL levels were increased in G4 (54.7%), followed by G5 (50.3%) and G3 (48.9%) (Table 5). The data show the glucose and lipid lowering potential of a T3 diet when fed to G4 rats.

The blood glucose and lipid levels of rats were totally changed following the consumption of RS3. The microscopic observation of the livers of the rats is shown in Figure 11. The tissue staining in the control showed the normal architecture of the liver with normal hepatocytes and a sinusoidal layer. In the diabetic control, due to the induction of diabetes, the liver showed the hepatocytes degeneration and the inflammatory cellular infiltration with fatty liver. However, in the treatment controls, the liver showed the mild degeneration of hepatocytes. The rat group (G4) who consumed T3 basmati rice showed less degeneration when compared to the group who consumed basmati rice stored with T1. On the other hand, while observing the pancreas, the tissue staining in the control group showed the normal architecture of the pancreas with a normal range of Langerhans islands and beta cells (Figure 12). The section of the pancreas of a diabetic rat depicted the cytoplasmic degeneration of islets of Langerhans. While Group 3 and 4 (treatment groups) showed a reduction in the number of beta cells and degranulated cytoplasm in most cells when compared to the control group, the results were far better than for the diabetic group. Group 4, in which basmati rice stored at 4 °C for 24 h was given provided, showed improved architecture of the pancreas, with increments in the formation of beta cells and the regeneration of islets of Langerhans when compared to group 3. Results revealed that the consumption of high RS rice products resulted in better insulin sensitivity and delayed glucose absorption, leading to improvements in blood glucose levels. Similarly, faecal bile excretion and reduced cholesterol synthesis via SCFA were the main factor controlling the triglyceride and cholesterol levels in the experimental rats.

## 5. Discussion

The goal of the current study was to determine the ideal method of cooking and storage conditions to raise the resistant starch content of the basmati rice products often consumed in India. It was discovered that the boiling 1 approach had a higher protein content than the boiling 2 method. This might have happened because of the effect of soaking and cooking, as in the boiling 2 method, cooking denatures protein and soaking solubilises protein content, leading to its reduction during the draining of water after cooking [25]. In a study, boiling was found to be superior (4.99%) among other heat treatments, like microwave (2.49%) and autoclaving (3.5%), increasing the protein content the most; this is due to the kinetic energy being mainly responsible for the increasing or decreasing of the protein content of rice while cooking [26].

Fat is important to increase the palatability of the food. Fried rice had a higher fat content because of the addition of extra oil and more absorption. When compared to frying, other cooking methods like boiling 1, boiling 2, and pressure cooking were found to provide a low fat content due to fat hydrolysis, which occurs in the presence of water and heat. The ash content was found to be higher in the boiling 1 and frying methods when compared to the boiling 2 and pressure cooking methods because of the higher amounts of protein, fat, and fibre content [27]. Boiling 2 had a low ash content; this could be because of macro and micro elements being leached out during soaking and draining, while the reduction in the fibre content could be a reason for the lower ash content observed in the pressure cooked rice sample.

Human digestive enzymes cannot break down dietary fibre. In addition to managing large intestine functions, this has a major physiological impact on glucose, lipid, and mineral metabolism [28]. The solubility of dietary fibre in water is primarily determined by its soluble or insoluble nature. Like a magnet, soluble dietary fibre attracts body water into the digestive tract [29]. Soluble fibre in the stomach creates a gel-like structure that is water soluble and facilitates easy food movement [30]. Water-insoluble fibres include lignin, cellulose, hemicellulose, and resistant starch. These fibres do not dissolve in water. Insoluble fibre helps move faeces out of the body by raising the intestinal pressure [31]. With the exception of frying, the amount of soluble dietary fibre in each cooking method was found to be similar in the current study. After cooking, the amount of soluble dietary fibre increased, but this is related to the temperature and cooking time. The primary cause of T1′s higher soluble fibre content when compared to T3 is the heat treatment of cereals, which increases the viscosities of the water extract, thereby converting insoluble dietary fibre into a soluble form. Boiling methods were shown to have significant amounts of insoluble dietary fibre. Maillard’s reaction is responsible for the rise in the amount of insoluble dietary fibre that occurs during cooking. It is possible that thermal processing may have caused the production of Maillard’s reaction products, and thus may have increased its IDF value [32]. An increase in the total dietary fibre content might have occurred due to the apparent increase in cellulose content [33]. The boiling or microwave heating of instant mashed potatoes has been shown to increase TDF and IDF contents [34]. Another study found that microwave cooking and frying reduced the in vitro digestible starch, and increased the resistant starch (RS) and water-insoluble dietary fibre (IDF) [35]. The heat processing of rice increased the dietary fibre content [36,37]. Hence, an increase in the cellulose content can lead to an increased amount of TDF and IDF. When compared to storage temperatures, a low temperature might be the cause of the increase in the amount of resistant starch in T3, resulting in higher levels of insoluble and total dietary fibre.

Resistant starch (RS) is defined as starch that is not digested by hydrolytic enzymes in the small intestine for at least 120 min after digestion, instead passing to the colon to then be fermented by microbiota [38]. It helps to control cholesterol, blood sugar levels, and essentially acts as an insoluble dietary fibre in the body [39]. Boiling as a cooking method will increase or decrease the resistant starch content, depending on the form of food [40]. High levels of RS were found in the boiling 1 method because of the presence of heat and water. Starch granules absorb moisture, swell, and gelatinise when food is heated in the presence of water, whereas amylose degrades and washes out of the solution to then produce a larger degree of gelatinisation with longer heating times [41,42]. In the boiling 2 method, the starch was leached out of rice and discarded with the water. Hence, a smaller amount of starch was available to convert into resistant starch, probably due to the low amount of amylose content. Fried rice contained high amounts of RS because stir-frying reduces the starch hydrolysis rate and increases the RS content. According to a study, stir-fried food’s RS concentration increased from 7 to 12% [43]. As the cooking time increased, so, it seemed, did the reductions in moisture content and amylose. The absence of water in the fried samples also inhibits the process of the crystallization of amylose chains, resulting in a decreased RS content [22,44].

The pressure cooked basmati rice had a significantly lower level of resistant starch than any other cooking processes because of the bursting of the starch cells that occurs at high temperatures and pressures. The mechanism of the high pressure gelatinisation of starch is different from heat-induced gelatinisation. The amorphous region of starch granules swells when coming into contact with water and high temperatures, leading to helix-coil transitions in amylose and amylopectin and a loss of granular structure and crystalline order [45]. Pressure cooking significantly reduced the RS content of fresh jasmine rice when compared to the rice cooker and oven baking [46]. Among storage temperatures, the RS content was to be found higher in T3 than in T2, T4, and T1. Because of the retrogradation, the recrystallization of starch granules occurred while storing at 4 °C. The rate and amount of retrogradation are determined by starch properties, like the molecular or crystalline structure, and storage conditions, such as time, temperature, and the water content [10].

Since retrogradation at low temperatures produces RS, insoluble dietary fibre, a high amylose concentration, and a longer amylopectin chain, it was discovered that the in vitro starch digestion rate of basmati rice products was lowest in T3 products, followed by T2, T4, and T1. A high amylose content reduced the starch digestibility because of the presence of large numbers of hydrogen bonds. The starch digestibility was also found to be low in fried rice. This might also be due to an interaction between amylose and fatty acids, resulting in complex formations on the surface of starch granules, and these components may act as physical barriers to digestion [47].

The starch digestion rate is calculated by taking the glucose measurements up to 180 min from the start of digestion, as rapidly digestible starch is digested at 20 min, slowly digestible starch is digested at 120 min, and complete starch digestion will have occurred after 180 min [48]. Rapidly digestible starch (RDS) is converted into glucose within 20 min of food intake, while slowly digestible starch (SDS) takes 20 to 120 min to be converted into glucose. Rapidly digestible starches are thought to cause an abrupt rise in blood glucose levels following consumption, whereas slowly digestible starches are completely digested in the small intestine, resulting in a slower rise in blood glucose. Slowly digestible starches are linked to satiety, a stable glucose metabolism, and diabetes management [49].

The RDS was found to be significantly high in pressure cooked basmati rice, followed by boiling 2, boiling 1, and fried rice. RDS was found to be high with T1, followed by T4, T2, and T3. SDS was found to be high in the boiling 1 method, followed by frying, boiling 2, and pressure cooking. The low digestibility could be caused by variations in the morphological and physical characteristics of the starches in cereals. Cooking cereal starches causes the starch granules to gelatinise and undergo physical and chemical disruption. In turn, the amount of water present, the cooking time, and temperature all affect the extent to which gelatinisation occurs [50]. Protein structures surrounding starch granules may limit starch gelatinisation and granule swelling, lessening the granules’ vulnerability to enzymatic attack. This may be responsible for a certain amount of the low digestibility. The type of starch has a direct effect on the rate at which cereal starches can be digested. The digestibility of starch decreases with increasing amylose concentration. The greater surface area of amylopectin and the insoluble aggregates are considered to be the cause of the apparent variations in the digestibility between amylose and amylopectin. These factors also probably reduce the susceptibility of cleavage sites to enzyme attack [51]. Therefore, the kind and supply of starch in cereals may affect how easily they digest.

Similarly, the amylose content in basmati rice was found to be high using the boiling 1 method when stored with T3, which could be because of how the amylose aligns and associates with itself during cooking and cooling, a process known as retrogradation [52]. The cooking process decreased the amylose content, probably due to its leaching into the water, thus resulting in a low amylose content following the boiling 2 method [53].

The rate at which a particular food raises the blood sugar is determined by its glycaemic index. Selecting low-glycaemic index foods has been shown to improve a healthy person’s post-prandial glucose and lipid metabolism. The glycaemic load (GL) of an average portion of food is made up by the amount of available carbohydrate in that serving and the GI of the food. The higher the GL, the greater the expected elevation in blood glucose and in the insulinogenic effect of the food. The consumption of a high-GL diet for a longer period of time leads to an increased risk of type 2 diabetes and coronary heart disease [54].

The glycaemic index and glycaemic load of the human subjects fed with basmati rice prepared with the boiling 1 method were found to be low with T3, followed by T2, which can be due to the higher amount of RS and dietary fibre found in these products. The result of the present study was supported by a randomised, single-blind crossover study, where 15 healthy participants were fed white rice cooked and stored at different temperatures. The findings showed that the white rice cooked and stored at 4 °C for 24 h had the highest RS content and also led to a reduced glycaemic response when compared to the control rice [55]. Following the administration of the treatment diet, the blood glucose levels of the pre-treatment groups (G3–G5) in the rat research dramatically restored to a near-normal range. G4 showed a significant fall in blood glucose levels—22.5%. Products made from rice that were kept with T3 were rich in resistant starch, which causes the sugar to be released into the bloodstream gradually and decreases the absorption of the sugar. The control of the glucose output is significantly impacted by the slow digestion of resistant starch (RS) [56]. The starch is nearly freshly digested, but the metabolism of resistant starch can occur anywhere from 5 to 7 h after a meal. Digestion takes 5–7 h, gradually raises the blood sugar levels, reduces blood sugar and insulinemia, and leads to satiety being provided for a longer time. Because insoluble dietary fibre absorbs sugar molecules, it inhibits the passage of glucose in the small intestine [57]. In the human digestive tract, fibre slows the rise in blood glucose levels and decreases the absorption of glucose. Also, when fibre is hydrated, it acts more effectively to reduce blood glucose level [58].

Similar results were reported in a study where rats fed on resistant starch from the rice group (12.9 ± 3.2 mmoles/L) exhibited a decreasing tendency in blood glucose concentrations when compared to the resistant starch of corn (15.6 ± 8.1 mmoles/L [59]. The relative body weight in the treatment group and diabetic group (G2) was significantly lower (<0.001) than that of the normal control (G1) group; this was due to the protein breakdown during diabetes. In diabetic control rats, the plasma insulin levels were found to be lower (G2) when compared to the normal control group (G1). However, the level of plasma insulin was found to be elevated in the treatment groups (G3–G5) when compared to the diabetic control group (G2) after the treatment diet was provided. STZ (streptozotocin) induces diabetes in rats via the preferential accumulation of the chemical in β-cells following entry through the GLUT2 (glucose transporter) receptor (chemical structural similarly with glucose), which allows STZ to bind to this receptor. Under this condition, the destruction of β-cells and the induction of the hyperglycaemic state are associated with inflammatory infiltrates, including lymphocytes in the pancreatic islets. The destruction of the β-cells of the islets of Langerhans results in a massive reduction of insulin release. However, in the treatment group, the plasma insulin levels increased significantly. This could happen because of the release of insulin in its bound form from the beta cells of islets of Langerhans [60].

After 28 days of treatment, the cholesterol level reduced significantly in all treatment groups, except for the diabetic control (G2). Insulin inhibits the lipolysis process because it is important to synthesise fatty acids and triglycerides in fat tissues. In the diabetic group, due to the reduction in insulin levels, the body of the rat started to use fat for the production of energy through the lipolysis mechanism [6], hence, resulting in the increased production of Acetyl-coA, which further increased the level of ketone bodies and cholesterol levels [61].

The diabetic group exhibited lower levels of high density lipoproteins (HDLs) and higher levels of total cholesterol, triglycerides, and LDL in comparison to the normal control and treatment groups. The maximum reduction in triglyceride was found in G4, when compared to other groups. These results may be related to soluble fibre consumption, which is considered an important dietary factor in the prevention of cardiovascular diseases in many epidemiological studies [62].

Interestingly, the results indicate that, at the end of the treatment, all groups of diabetic rats treated with basmati rice products had lower cholesterol levels, lower triglyceride and LDL levels, and higher HDL levels. The amount of resistant starch found in rice in each diet has an impact on this outcome. The highest concentrations of resistant starch demonstrated their ability to lower the LDL, triglyceride, and total cholesterol levels. Diabetes is abnormally linked to increased amounts of endogenous glucose and fat synthesis. Through the regulation of glycolysis and gluconeogenesis, the liver plays a critical role in preserving glucose utilisation and storage equilibrium [63].

Microscopic observations of the liver cells of rats showed that the group that consumed T3 basmati rice exhibited less degeneration when compared to the T1 group, which could be due to the presence of a higher amount of resistant starch in T3 basmati rice, exhibiting better insulin sensitivity and an improved lipolysis mechanism. Retrograded starch has also been suggested to reduce blood LDL and cholesterol concentrations in a number of pathways, such as an increase in bile acid excretion from faeces. In addition to short-chain fatty acids in the liver and guts of rats, soluble dietary fibres regulate cholesterol levels by inhibiting hepatic cholesterol production through large intestine fermentations [64].

The histopathological study of pancreases from group 4 showed the improved architecture of the pancreas, with increments in the formation of beta cells and the regeneration of islets of Langerhans when compared to group 3, leading to improved blood glucose levels. High levels of RS, which function as an insoluble fibre and are fermented by the intestinal microbiota, are responsible for this. They also release carbon dioxide, methane, hydrogen, and metabolically active short-chain fatty acids, which have an impact on insulin secretion and hepatic gluconeogenesis [65]. According to a recent study, RS from a high-fat diet can control the expression of genes related to lipid and hepatic glucose metabolism pathways, as well as hyperglycaemia and hyperlipidaemia in diabetic rats. Resistant starch and insoluble dietary fibre also help to reduce lipolysis and increase GLP-1, peptide YY, and insulin secretion [66]. GLP-1 (Glucagon-like-peptide 1) stimulates insulin secretion and reduces glucagon secretion. Pancreatic peptides (YY) help to reduce appetite and increase the feeling of fullness, which then helps to control blood glucose and lipid levels.

## 6. Conclusions

The quantity of resistant starch in basmati rice increased when cooking methods such as boiling 1, boiling 2, and frying were used, while the amount dropped when pressure cooking was used. While freshly prepared food (T1) and reheated food (T4) reduced the amount of resistant starch, products held at 4 °C (T3) and at room temperature for 24 h (T2) increased the amount of resistant starch. Products kept at 4 °C (T3) exhibited high levels of amylose, slowly digestible starch, and insoluble dietary fibre. The trial rats’ blood glucose and cholesterol levels dropped as a result of eating this rice, and the human participants’ glycaemic index and glycaemic load also fell. A slower rise in the blood glucose and cholesterol levels was also caused by the food product’s greater RS, which stimulated the regeneration of beta cells in the pancreas and hepato cells in the liver. Given the wide range of starchy preparations that Indians eat, modifying the method of cooking and storage temperature of starchy foods may have multiple positive health effects. In order to manage blood glucose levels, awareness regarding the nutritional and health benefits of consuming resistant starch should be raised. In order to boost the amount of resistant starch in food at the domestic level, people must also be educated on proper food preparation and storage techniques.

## Figures and Tables

**Figure 1 foods-13-01669-f001:**
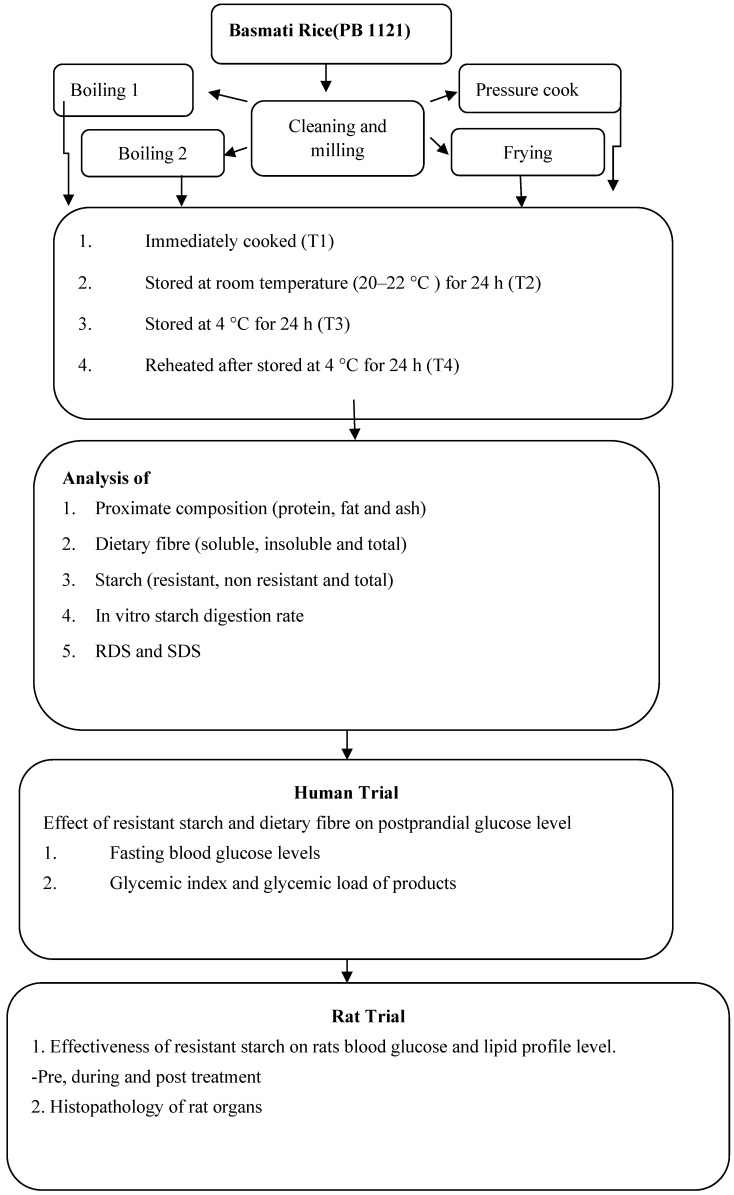
Methodology of the study.

**Figure 2 foods-13-01669-f002:**
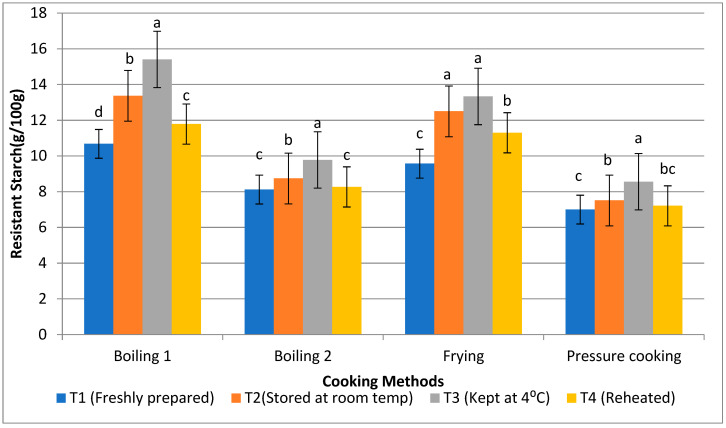
Effect of different cooking methods and storage temperatures on the resistant starch content of basmati rice products (g/100 g). Data are means ± SD. Different letters over the error bars denote that the means differed significantly (*p* < 0.05).

**Figure 3 foods-13-01669-f003:**
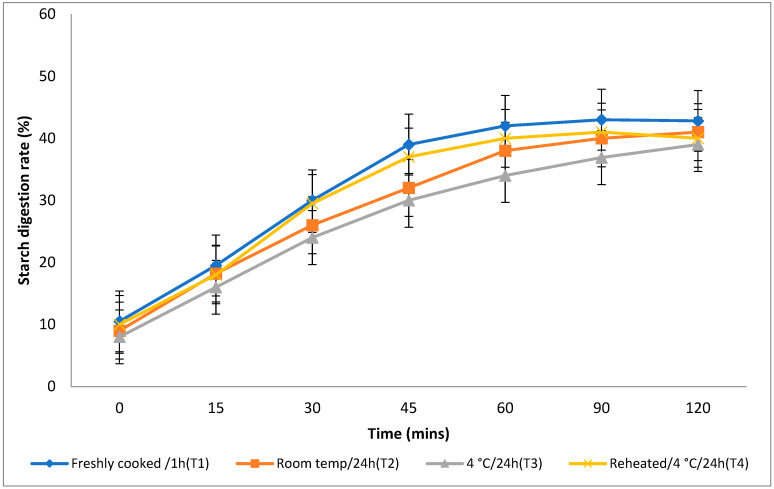
Effect of different storage temperatures on the in vitro starch digestion rate of basmati rice boiled via absorption.

**Figure 4 foods-13-01669-f004:**
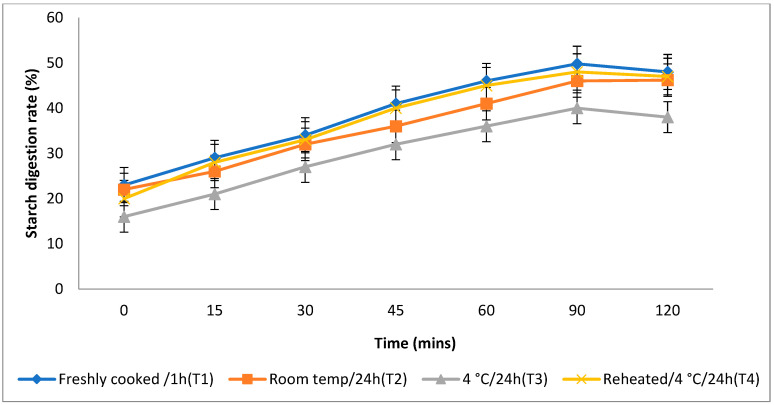
Effect of different storage temperatures on the in vitro starch digestion rate of basmati rice boiled in an extra amount of water.

**Figure 5 foods-13-01669-f005:**
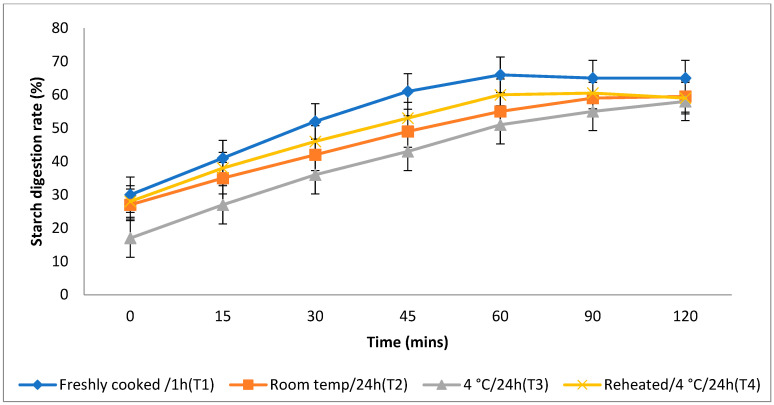
Effect of different storage temperatures on the in vitro starch digestion rate of pressure cooked basmati rice.

**Figure 6 foods-13-01669-f006:**
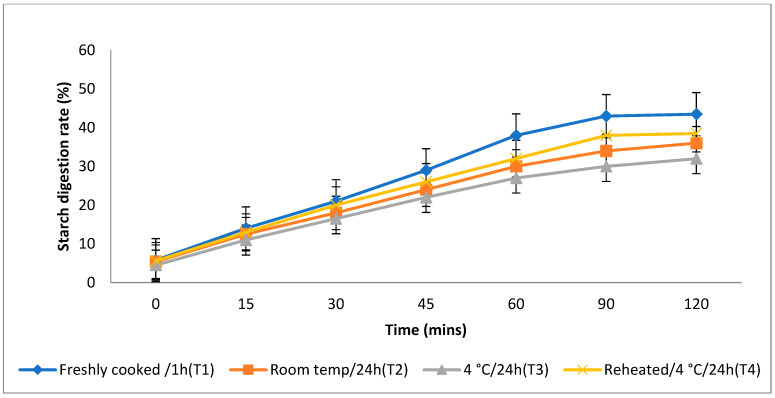
Effect of different storage temperatures on the in vitro starch digestion rate of fried basmati rice.

**Figure 7 foods-13-01669-f007:**
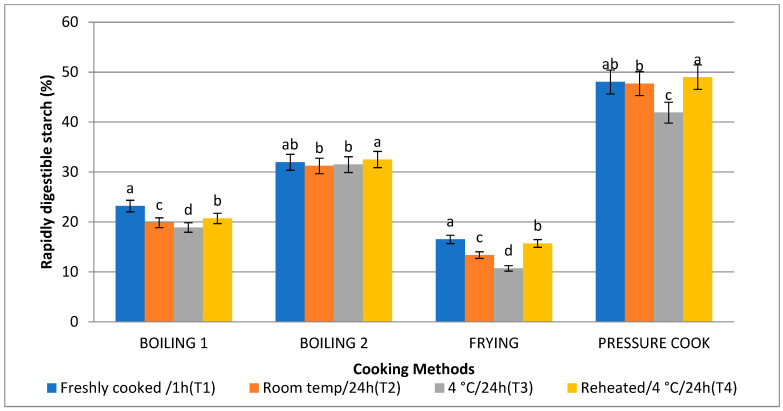
Effect of different cooking methods and storage temperatures on rapidly digestible starch in basmati rice products. Data are means ± SD. Different letters over the error bars denote that the means differed significantly (*p* < 0.05).

**Figure 8 foods-13-01669-f008:**
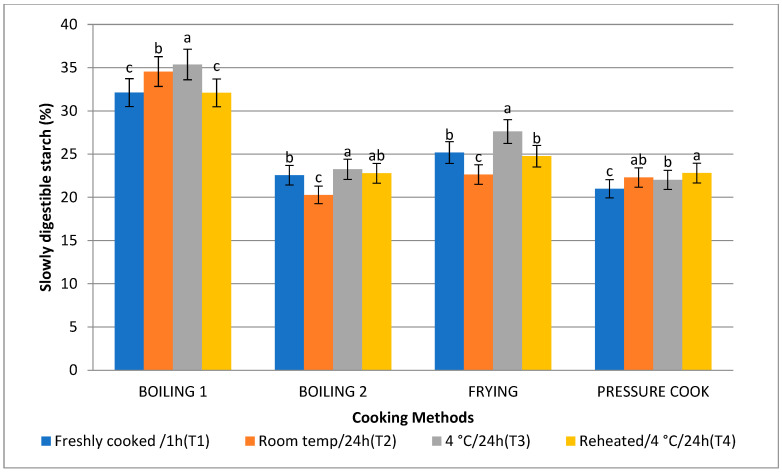
Effect of different cooking methods and storage temperatures on slowly digestible starch in basmati rice products. Data are means ± SD. Different letters over the error bars denote that the means differed significantly (*p* < 0.05).

**Figure 9 foods-13-01669-f009:**
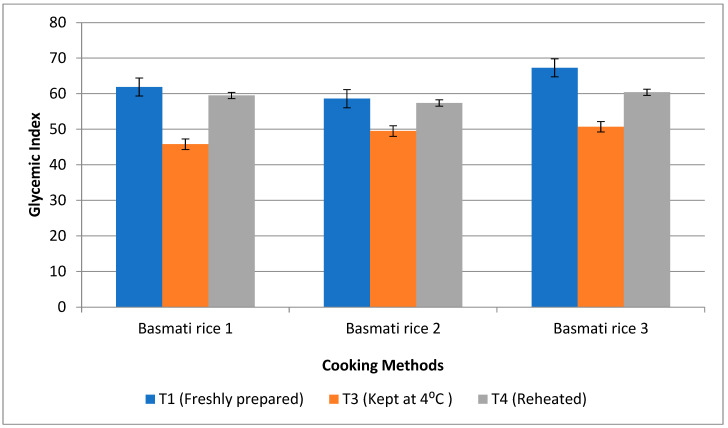
Effect of different cooking methods and storage temperatures on the glycaemic indexes of basmati rice products.

**Figure 10 foods-13-01669-f010:**
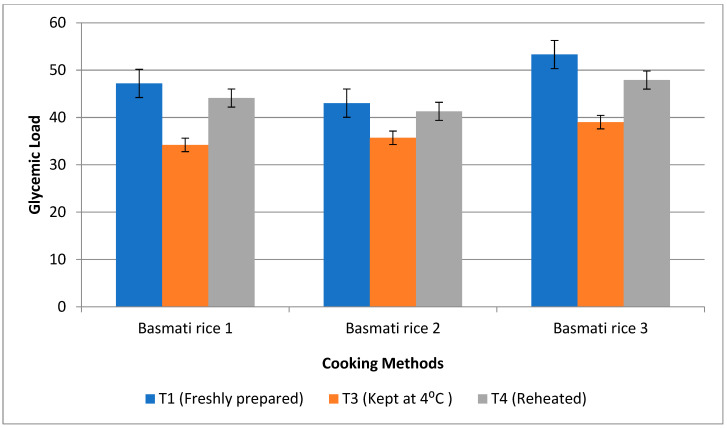
Effect of different cooking methods and storage temperatures on the glycaemic loads of basmati rice products.

**Figure 11 foods-13-01669-f011:**
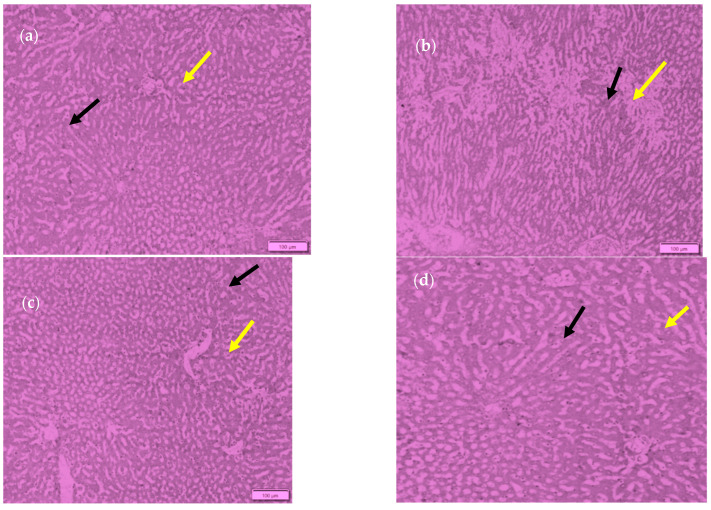
Histopathology study of the livers of rats. (**a**) Control; (**b**) diabetic control; (**c**) freshly prepared basmati rice-fed diet group; (**d**) T3 basmati rice-fed diet group. Black arrow shows hepatocytes; yellow arrow shows the sinusoidal layer.

**Figure 12 foods-13-01669-f012:**
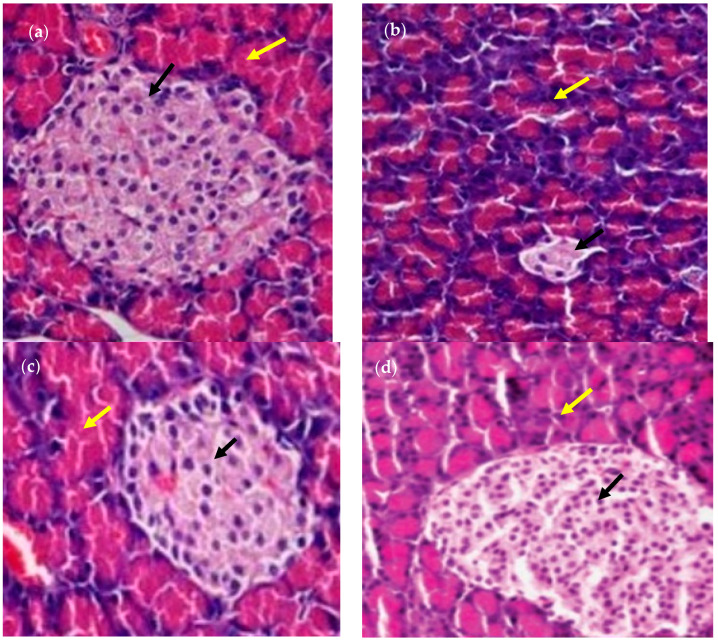
Histopathology study of the pancreas of rats. (**a**) Control; (**b**) diabetic control; (**c**) T1 basmati rice-fed diet group; (**d**) T3 basmati rice-fed diet group. Black arrows show beta cell; yellow arrows show Langerhans islet.

**Table 1 foods-13-01669-t001:** Preparation of commonly consumed basmati rice products in India.

Cereal Product	Ratio of Ingredients	Method
Boiling 1 (rice was boiled in an equal amount of water i.e., boiling by absorption)	Basmati rice to water (2:4 *w*/*v*)	Basmati rice was boiled in water for 15 min.
Boiling 2 (after boiling, extra water was drained out)	Basmati rice to water (2:7–8 *w*/*v*)	Basmati rice was boiled in an extra amount of water separately for 15 min, and then the extra water was drained out after cooking.
Frying (fried rice)	Basmati rice to water (2:4 *w*/*v*)	Wok was heated on a high flame and 10 mL of oil was used to fry rice. Then the rice was cooked for about 20–25 min.
Pressure cooking (pressure cooked rice)	Basmati rice to water (2:4 *w*/*v*)	Basmati rice was pressure cooked separately in a pressure cook (15 psi) for 10 min.

**Table 2 foods-13-01669-t002:** Effects of different cooking methods and storage temperatures on the crude protein, crude fat, and ash content of basmati rice products (g/100 g).

Cooking Methods	Treatments				
	T1	T2	T3	T4	Treatment Mean
Crude protein					
Boiling 1	8.68 ± 0.22 ^b^	9.32 ± 0.57 ^b^	10.44 ± 0.19 ^b^	9.74 ± 0.52 ^a^	9.54 ^A^
Boiling 2	9.51 ± 0.36 ^b^	7.62 ± 0.07 ^c^	9.90 ± 0.10 ^b^	6.87 ± 0.42 ^cd^	8.22 ^B^
Frying	7.59 ± 0.10 ^c^	8.80 ± 0.10 ^b^	9.34 ± 0.14 ^c^	7.40 ± 0.46 ^b^	8.28 ^B^
Pressure cooking	6.34 ± 0.44 ^de^	7.54 ± 0.43 ^a^	8.21 ± 0.26 ^cd^	5.57 ± 0.20 ^e^	7.49 ^C^
Storage Mean	8.03 ^B^	8.32 ^B^	9.47 ^A^	7.39 ^C^	
Crude fat					
Boiling 1	2.01 ± 0.04 ^b^	2.37 ± 0.15 ^b^	2.47 ± 0.07 ^b^	2.38 ± 0.83 ^b^	2.18 ^B^
Boiling 2	1.91 ± 0.08 ^b^	1.84 ± 0.07 ^bc^	1.93 ± 0.06 ^b^	1.95 ± 0.05 ^b^	1.90 ^B^
Frying	10.60 ± 0.40 ^a^	10.98 ± 0.49 ^a^	11.31 ± 0.30 ^a^	10.57 ± 0.64 ^a^	10.86 ^A^
Pressure cooking	0.62 ± 0.05 ^d^	0.83 ± 0.14 ^d^	0.90 ± 0.10 ^cd^	0.80 ± 0.10 ^d^	0.78 ^C^
Storage Mean	3.78 ^A^	4.0 ^A^	4.03 ^A^	3.92 ^A^	
Ash					
Boiling 1	1.79 ± 0.09 ^ab^	1.82 ± 0.04 ^a^	1.90 ± 0.03 ^a^	1.95 ± 0.05 ^a^	1.86 ^A^
Boiling 2	1.72 ± 0.14 ^ab^	1.71 ± 0.05 ^abc^	1.87 ± 0.03 ^a^	1.71 ± 0.11 ^abc^	1.75 ^B^
Frying	1.89 ± 0.01 ^a^	1.92 ± 0.07 ^a^	1.85 ± 0.06 ^a^	1.74 ± 0.21 ^ab^	1.85 ^AB^
Pressure cooking	1.43 ± 0.10 ^cd^	1.53 ± 0.13 ^bc^	1.22 ± 0.12 ^d^	1.69 ± 0.08 ^abc^	1.46 ^C^
Storage Mean	1.70 ^A^	1.74 ^A^	1.70 ^A^	1.77 ^A^	

Values are mean ± SD; Mean values with different superscripts are significantly (*p* ≤ 0.05) different. **T1**—Freshly prepared with in 1 h, **T2**—Stored at room temperature (20–22 °C for 24 h), **T3**—Kept at 4 °C for 24 h, **T4**—Reheated samples after stored at 4 °C for 24 h. Boiling 1 (rice boiled by absorption); boiling 2 (rice boiled in an extra amount of water); frying (fried rice); pressure cooking (rice cooked in pressure cooker).

**Table 3 foods-13-01669-t003:** Effect of different cooking methods and storage temperatures on the dietary fibre (soluble, insoluble, and total) content of basmati rice products (g/100 g).

Cooking Methods	Treatments				
	T1	T2	T3	T4	Treatment Mean
Soluble dietary fibre					
Boiling 1	1.10 ± 0.10 ^ab^	0.89 ± 0.09 ^bcde^	0.59 ± 0.02 ^fg^	0.93 ± 0.06 ^bcd^	0.87 ^A^
Boiling 2	1.16 ± 0.06 ^a^	0.80 ± 0.00 ^cdef^	0.63 ± 0.06 ^fg^	0.97 ± 0.06 ^abcd^	0.89 ^A^
Frying	0.89 ± 0.10 ^bcde^	0.77 ± 0.03 ^def^	0.55 ± 0.05 ^g^	0.90 ± 0.00 ^bcd^	0.78 ^B^
Pressure cooking	1.17 ± 0.15 ^a^	0.87 ± 0.06 ^cde^	0.68 ± 0.09 ^efg^	0.99 ± 0.01 ^abc^	0.92 ^A^
Storage Mean	1.08 ^A^	0.83 ^C^	0.61 ^D^	0.94 ^B^	
Insoluble dietary fibre					
Boiling 1	2.43 ± 0.06 ^defg^	3.10 ± 0.10 ^ab^	3.33 ± 0.12 ^a^	2.33 ± 0.15 ^defgh^	2.80 ^A^
Boiling 2	2.23 ± 0.06 ^fghi^	2.50 ± 0.00 ^cdef^	2.87 ± 0.06 ^bc^	2.30 ± 0.10 ^efgh^	2.47 ^B^
Frying	1.92 ± 0.07 ^i^	2.50 ± 0.24 ^def^	2.62 ± 0.26 ^cde^	1.99 ± 0.01 ^hi^	2.25 ^C^
Pressure cooking	2.00 ± 0.10 ^hi^	2.36 ± 0.14 ^defgh^	2.70 ± 0.10 ^cd^	2.07 ± 0.06 ^ghi^	2.28 ^C^
Storage Mean	2.14 ^C^	2.61 ^B^	2.88 ^A^	2.17 ^C^	
Total dietary fibre					
Boiling 1	3.53 ± 0.15 ^bc^	3.99 ± 0.01 ^a^	3.92 ± 0.14 ^ab^	3.27 ± 0.15 ^cde^	3.67 ^A^
Boiling 2	3.40 ± 0.10 ^cd^	3.30 ± 0.00 ^cd^	3.50 ± 0.10 ^c^	3.27 ± 0.06 ^cde^	3.36 ^B^
Frying	2.82 ± 0.16 ^f^	3.27 ± 0.21 ^cde^	3.17 ± 0.21 ^cdef^	2.89 ± 0.01 ^ef^	3.03 ^D^
Pressure cooking	3.17 ± 0.21 ^cdef^	3.23 ± 0.13 ^cde^	3.38 ± 0.16 ^cd^	3.06 ± 0.05 ^def^	3.20 ^C^
Storage Mean	3.22 ^B^	3.44 ^A^	3.49 ^A^	3.12 ^B^	

Values are mean ± SD; Mean values with different superscripts are significantly (*p* ≤ 0.05) different. **T1**—Freshly prepared with in 1 h, **T2**—Stored at room temperature (20–22 °C for 24 h), **T3**—Kept at 4 °C for 24 h, **T4**—Reheated samples after stored at 4 °C for 24 h. Boiling 1 (Rice boiled by absorption); boiling 2 (Rice boiled in an extra amount of water); frying (Fried rice); pressure cooking (Rice cooked in pressure cooker).

**Table 4 foods-13-01669-t004:** Effectiveness of resistant starch on blood glucose and plasma insulin levels in rats.

	Blood Glucose Level (mg/dL)				Plasma Insulin (µ/mL)		
Diet Group	Pre Treatment	During Treatment	Post Treatment	Treatment Mean	Initial (Pre)	Final (Post)	*p* Value
G1	114.3 ± 6.86 ^g^	115 ± 5.4 ^g^	112.6 ± 4.32 ^g^	114.0 ^E^	24.33 ± 0.82	24.833 ± 1.17 ^a^	0.36 ^NS^
G2	286.5 ± 16.5 ^a^	285.6 ± 13.8 ^a^	283.3 ± 8.35 ^a^	285.1 ^A^	12.67 ± 1.63	10.00 ± 0.89 ^d^	≤0.001 *
G3	286.3 ± 16.4 ^a^	268 ± 15.9 ^abcd^	244.2 ± 11.0 ^bcde^	266.1 ^B^	12.00 ± 1.26	16.833 ± 1.72 ^c^	0.005 *
G4	282.1 ± 18.4 ^ab^	238.17 ± 19.4 ^cde^	198.3 ± 18.1 ^f^	239.5 ^D^	13.00 ± 1.79	21.500 ± 0.54 ^b^	≤0.001 *
G5	276.6 ± 30.1 ^abc^	257.7 ± 28.4 ^abcde^	232.5 ± 25.8 ^def^	255.6 ^BC^	12.33 ± 1.03	17.167 ± 0.75 ^c^	≤0.001 *

Each value is the mean of six observations. Values are mean ± SD; Mean values with different superscripts are significantly (*p* ≤ 0.05) different. NS (non-significant). Table showed the standard of the mean of diet groups; 1. Control group (normal diet-fed rats) (G1); 2. Diabetic control (Normal diet-fed rats) (G2); 3. Freshly prepared within 1 h basmati rice-fed rats (G3); 4. Stored at 4 °C for 24 h basmati rice stored fed-rats (G4); 5. Reheated after being stored at 4 °C for 24 h basmati rice-fed rats (G5). * Significant at 5%.

**Table 5 foods-13-01669-t005:** Effect of resistant starch on the lipid profile of rats.

	Total Cholesterol (mg/dL)			Triglyceride (mg/dL)			HDL (High Density Lipoprotein) (mg/dL)			LDL (Low Density Lipoprotein)(mg/dL)		
Diet Group	Initial (Pre)	Final (Post)	*p* Value	Initial (Pre)	Final (Post)	*p* Value	Initial (Pre)	Final (Post)	*p* Value	Initial (Pre)	Final (Post)	*p* Value
**G1**	85.17 ± 3.97	85.500 ± 2.58 ^cd^	0.805 ^NS^	72.33 ± 5.96	73.833 ± 6.145 ^d^	0.122 ^NS^	54.16 ± 4.57	56.167 ± 1.16 ^a^	0.356 ^NS^	23.5 ± 1.048	24.333 ± 1.032 ^f^	0.201 ^NS^
**G2**	137.3 ± 11.05	165.50 ± 05.57 ^a^	0.003 *	126.83 ± 1.47	124.50 ± 02.66 ^a^	0.084 ^NS^	24.66 ± 1.36	23.333 ± 1.21 ^e^	0.034 *	74.33 ± 1.632	77.500 ± 3.78 ^a^	0.141 ^NS^
**G3**	134.83 ± 7.22	91.000 ± 0.89 ^b^	≤0.001 *	129.66 ± 3.66	103.167 ± 1.47 ^b^	≤0.001 *	23.5 ± 2.58	46.000 ± 0.89 ^c^	≤0.001 *	70.5 ± 2.88	62.500 ± 0.54 ^b^	0.019 *
**G4**	138.33 ± 10.74	83.000 ± 0.89 ^d^	≤0.001 *	128.16 ± 1.72	88.000 ± 1.89 ^c^	≤0.001 *	24.66 ± 1.86	54.500 ± 1.64 ^a^	≤0.001 *	68.5 ± 2.25	47.667 ± 2.06 ^e^	≤0.001 *
**G5**	134.0 ± 8.31	89.000 ± 0.89 ^bc^	≤0.001 *	125.83 ± 3.76	99.333 ± 1.50 ^b^	≤0.001 *	23.66 ± 1.21	47.500 ± 0.54 ^c^	≤0.001 *	68.66 ± 2.65	56.500 ± 1.76 ^c^	≤0.001 *
		180–200		<150			>50			<100		

Each value is the mean of six observations. Values are mean ± SD; Different small letters in different columns showed significant differences at 5%; * Significant at 5%; NS non-significant. Table showed the standard of the mean of diet groups; 1. Control group (normal diet-fed rats) (G1); 2. Diabetic control (Normal diet-fed rats) (G2); 3. Freshly prepared with in 1 h basmati rice-fed rats (G3); 4. Stored at 4 °C for 24 h basmati rice-fed rats (G4); 5. Reheated after being stored at 4 °C for 24 h basmati rice fed rats (G5).

## Data Availability

The original contributions presented in the study are included in the article, further inquiries can be directed to the corresponding author.

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
