# Peer review of "Analysing the Impact of Resistant Starch Formation in Basmati Rice Products: Exploring Associations with Blood Glucose and Lipid Profiles across Various Cooking and Storage Conditions In Vivo"

_foods, 2024, doi:10.3390/foods13111669_

Round 1
Reviewer 1 Report
Comments and Suggestions for Authors
The basmati rice products were prepared with four cooking methods and four storage methods, and evaluated the effects on blood glucose level and lipid profile in humans and rats. The study is very interesting.
1) The writing version of this paper should be checked again and again, the punctuation and space should be checked. For example, Line 29-30, Thus helps to inhibit glycogen and lipid synthesis.C;
2) Figure 1 is not complete, and is not easy to read. 920-22℃should be checked. The design of human trial in line 201-211 should be described more detailedly.
3)The data in figures 2, 7 and 8 are not clear.
4) Table 5 is not complete.
5) The plates in figure 15 are illusory and dreamlike to look at , not clear.
6) Boiling 1 method was titled as boiling by absorption, it is not easy to understand.
7) Figure 9-10 did not have titles for Y-axis.
8) The data in tables 4-5 did not have the number after the decimal point.
9) Line 552-559, on rice retrogradation, the authors should suggest that in India and south-east Asia, people like stale rice, but in China and north-east Asia, people like fresh rice and use different methods to inhibit starch retrogradation. The taste and texture of cooking rice with high resistant starch should be discussed.
10)Line 179, Check C2H3NaO2. The temperature accuracy in water bath should be given.
12)Line 110, When was the Basmatic rice sample harvested from the field ? What was its moisture content after being milled? How to store the milled rice during experiments?
13) Line 226-227, how to make the rats to completely eat sixteen types of rice? The number of parallel tests should be given.
14) In the discussion section, for the cooked basmati rice with improved resistant starch, how did vitamins change?
15) In the 77 refs, 65 refs are before 2014, most of them should be renewed.
Author Response
- The writing version of paper( punctuation and space) has been checked properly. All the needful mistakes have been rectified.
- All the needful Corrections in figure 1 have been done along with temperature correction and completeness of figure.
- The data in figure 2 has been revised But figures 7 and 8 already have the correct data.
- Table 5 has been completed.
- I don't get the figure 15. Total number of figures is 12 in this paper.
- Boiling 1 by absorption (is the equal Amount of water to rice to boil them) has been added.
- Y- axis has been added in Figure 9-10.
- The number after the decimal point has been added in tables 4-5.
- As suggested in Line 552-559, on rice retrogradation, the authors have added that in India and south-east Asia, people like fresh rice, but in China and north-east Asia, people like stale rice.
- Line 179, the temperature of the water bath and C2H3NaO2 has been checked and they are correct.
- Line 110, needful, was mentioned. Basmati rice was harvested in the month of October from the field. The moisture content of basmati rice was reduced up to 12 percent after milling. The samples were dried and stored in zip lock bags (airtight packages) and further used for nutritional analysis after the treatments.
- Line 226-227, mentioned that each group contained 7 rats. Among four different types of cooking, only rice prepared by boiling 1 method was given to the rats as limited numbers of rats were available to us.
- The effect of improved resistant starch on vitamins has not been studied. So this part has not been added.
- About references, I tried my best to add the recent references related to study.
Please see the attachment

Reviewer 2 Report
Comments and Suggestions for Authors
Analyzing the Impact of Resistant Starch Formation in Basmati
Rice Products: Exploring Associations with Blood Glucose and
Lipid Profiles across Various Cooking and Storage Conditions
This article seems to be interesting and immediately useful. However, there are some points that require attention.
- In Figure 1, improve the arrows in the flowchart. There is a paragraph with an error: "Stored at room temperature 920-22⁰C) for 24h (T2)"
- As for the analytical tests, they would have greatly improved if glycated hemoglobin had also been analyzed in addition to liver markers.
- In Table 2, and in all tables and figures, correctly indicate the units (g/100 g).
- All figures must present error bars, as well as letters indicating the result of the ANOVA statistical treatment.
- This topic has been abundantly studied before. Please clearly indicate in the introduction section the novelty of the present study.
Author Response
- All the needful Corrections in figure 1 have been done along with temperature correction, completeness and arrows of figure.
- In the study, unfortunately, the glycated hemoglobin has not been studied. So data, related to it has not been added.
- All tables and figures correctly indicated the units (g/100 g).
- Error bars have been added in the figures.
- The introduction section has been revised and improved.
please see the attachment

Reviewer 3 Report
Comments and Suggestions for Authors
Overall, the document appears to do a good job of presenting the results in a manner that supports the conclusions drawn, provided that the data is accurately represented and critically analyzed in the context of existing literature and the study's limitations.
Title: The title is appropriate and specific, yet it could be slightly adjusted to highlight the focus on human health.
Abstract: Improve the clarity of the study's purpose in the abstract by specifying the importance of the research for public health or clinical nutrition. Include a brief mention of the methods used, the main results, and the practical implications or recommendations for consuming basmati rice in healthy diets.
The introduction in your article provides a good background, however, to further strengthen the introduction and ensure it provides sufficient background for the study, consider the following aspects:
1. Ensure the introduction includes a review of recent studies that highlight the current understanding and gaps in research regarding resistant starch, especially in basmati rice products. This could help to establish the relevance and timeliness of your study.
2. Clearly state the research gap your study intends to fill. While the introduction outlines the benefits of resistant starch, it could more explicitly articulate the need for research on basmati rice products under various cooking and storage conditions and their specific effects on health markers.
3. Conclude the introduction with a clear justification for your study and specific research objectives.
Material and methods
The document outlines various cooking methods and storage conditions, followed by nutritional analysis, in-vitro digestion rate assessments, and in-vivo studies on glycemic index and lipid profiles. These methods are appropriate for the study's goals, but the specificity of protocols, measurement techniques, and statistical analyses used would further validate the research design's rigor.
To ensure the research design is appropriate, consider:
1. Detailing how the subjects are allocated to experimental groups.
2. Describing the steps taken to minimize bias and ensure reliability and validity of the findings.
3. Explaining the rationale behind the choice of specific methods and measures.
Please review Figure 1, as a "9" was placed instead of the parenthesis, "920-22°C )".
The authors should review Figure 1, as there are sentences missing within the text boxes.
Author Response
- In vivo could be Added in title.e.g, Title: Analyzing the impact of Resistant Starch Formation in Basmati Rice Products: Exploring Associations with Blood Glucose and Lipid Profiles across various cooking and storage conditions in Vivo
- Abstract: has-been improved. Brief discussion about methods , recommendations and importance of research has been added. (Next, the products' proximate composition (AOAC 2000), total dietary fibre (Megazyme K-TDFR-200A kit), resistant starch (RS) megazyme K-RSTAR kit), (Awareness among people regarding the nutritional and health benefits of resistant starch consumption and education to people about the right way of cooking at domestic level by simple modifications in food processing methods and their storage temperatures can provide health advantages to control blood glucose and lipid profile from starchy foods).
- Introduction part has been improved. The research gap and objectives of study have been added. (The main challenge to the food industry is the manufacturing of consumer-friendly foods, which contains enough resistant starch to result in a great enhancement in public health. Though the health benefits provided by resistant starch have been well documented, few studies are available on the resistant starch content of basmati rice products in India and in vivo efficacy of resistant starch of basmati rice products to improve glucose and lipid profile have not been studied.
Objectives
- To determine the effect of cooking and refrigeration on the resistant starch and dietary fibre content of cereal products.
- To quantify the resistant starch and soluble fibre components in cereal products.
- To assess the impact of resistant starch and soluble fibre components on postprandial glucose response and appetite rating.
- To determine the effectiveness of resistant starch on lipid profile and blood glucose level in rat model.
- All the needful Corrections in figure 1 have been done along with temperature correction, completeness and arrows of figure.
Please see the attachment
